# Survey on barriers to psychiatrists' use of clozapine for young people in Scotland and suggestions for reducing these

**Graham Walker** [1,2]*, **Jason Lang**[1,2], **Helen Smith**[1,2]

**1** University of Glasgow, Glasgow, United Kingdom, **2** NHS Scotland, United Kingdom

* graham.walker@glasgow.ac.uk

## Abstract

### Introduction

The Mental Welfare Commission for Scotland published a report into the death of a young person, with recommendations for the Royal College of Psychiatry in Scotland Child and Adolescent Faculty; to explore if there were barriers to the use of Clozapine in young people in Scotland.

### Methods

A mixed-methods study was performed using a cross-sectional survey of clinicians working in child and adolescent psychiatry across Scotland, to determine attitudes towards clozapine use and the perceived barriers and facilitators to clozapine treatment.

### Results

Results suggest that there may be a lack of clearly defined pathways within and between services, as well as a lack of resources provided for the necessary monitoring of a young person started on clozapine. Multiple respondents felt unskilled in clozapine initiation and had not accessed formal training. The most frequently mentioned themes for improving facilitation of clozapine prescription were that of increased resources and training.

### Discussion

National policymakers including the Mental Welfare Commission, NHS Education for Scotland, and NHS Scotland should consider these findings to address the potential underutilisation of clozapine for people aged under 18 in services across Scotland. A review of current service provision should take place, with consideration of whether the facilitators to clozapine prescription which our study has highlighted could be implemented more effectively. This may help reduce identified barriers and increase clozapine prescription to those who would benefit from it, potentially improving outcomes for young people with treatment-resistant psychosis.

**Data Availability Statement:** All relevant data are within the manuscript.

**Funding:** The author(s) received no specific funding for this work.

**Competing interests:** The authors have declared that no competing interests exist.

## Introduction

The Mental Welfare Commission for Scotland (MWCFS) published a report in September 2023, looking into the death of a young person [1]. Mr D was treated for a psychotic disorder, which at times resembled a bipolar disorder but also featured symptoms more suggestive of schizophrenia (some clinicians would refer to this kind of presentation as schizoaffective disorder). Key findings of the report included that while it proved difficult to establish Mr D on an effective dose of antipsychotic medication of therapeutic benefit to him due to side effects and tolerability, his ongoing illness warranted the outlining of a clearer rationale for continuing, changing, or stopping medication and the effects of such changes. This could have assisted in decision-making about switching to treatment with the antipsychotic Clozapine. The report recommended that the Royal College of Psychiatry in Scotland (RCPsychiS) Child and Adolescent Faculty (CaAF) explore if there were barriers to the use of Clozapine in young people in Scotland [1].

Delays in the treatment of schizophrenia are associated with poorer outcomes; the longer the duration of untreated psychosis the worse the condition can become, with functional outcomes appearing to decline sharply. Early treatment has the potential to reduce the secondary impacts of this serious mental illness such as suicide, stigma, isolation and reduction in social status [2, 3]. National Institute for Health and Care Excellence (NICE) guidelines indicate that clozapine should be considered for young people with schizophrenia whose illness has not responded adequately to pharmacological treatment despite the sequential use of adequate doses of at least two different antipsychotic drugs each used for 6 to 8 weeks [4]. Of interest, the Scottish Intercollegiate Guidelines Network (SIGN) do not have a specific guideline for management of schizophrenia or psychotic illness in people under 18. SIGN Guideline 131: Management of Schizophrenia is the closest to such a guideline, but is focused on treatment of an adult population [5].

A 2022 systematic review concluded that clozapine use in childhood and adolescent onset schizophrenia was generally well tolerated and superior in efficacy to other antipsychotics. Clozapine showed superior efficacy over other antipsychotics in the short term (6 weeks) and long term (2–9 years), even in traditional antipsychotic-resistant cases [6]. It is suggested that clozapine is under prescribed as a treatment, in part due to the fact that both clinicians and service users may be concerned about, or unfamiliar with, its unusual potentially hazardous adverse effects [7]. A 2005 survey of all consultant psychiatrists working across adolescent units in the United Kingdom (UK) was put to 83 clinicians, of whom 59 responded (71%). More than 40% of respondents indicated that they do not prescribe clozapine, and the study concluded that those who do may not always be following best practice recommendations [2].

Given this evidence and the MWCFC recommendation, we investigated any barriers or facilitators relating to the use of Clozapine in children and young people in Scotland. This paper reports the results of a survey of the attitudes and prescribing practices related to clozapine in a sample of psychiatrists working in child and adolescent services within Scotland.

## Methods

We employed a mixed methods design, using a cross-sectional survey of psychiatrists working in child and adolescent psychiatry across Scotland, to determine attitudes towards clozapine use in CAMHS (Child and Adolescent Mental Health Services) and identify any perceived barriers or facilitators to clozapine treatment.

### Survey instrument

An online survey was developed by two authors (GW and HS) following guidelines for electronic survey design [8]. Previous research studies on clinician attitudes to clozapine were

considered during the design process [2, 9]. Prior to distributing the survey, it was presented to the RCPsychiS CaAF executive committee, with feedback used to further improve the final design. The survey consisted of ten questions. Questions one to three focused on the demographics of the psychiatrists completing the survey (grade, years of experience and area of work). Questions four and five focused on when the respondent last prescribed clozapine and potential barriers to prescription. Question six to eight focused on psychiatrists perceived skill levels with regards to clozapine prescription, training opportunities taken in this area and suggestions on factors that would help facilitate clozapine prescription. Questions nine and ten posed free text questions on the matter, including consideration of how to improve Scottish CAMHS psychiatrists' confidence in prescribing clozapine.

### Recruitment

The survey was distributed amongst psychiatrists within the RCPcyshiS CaAF. It was initially sent in a virtual form via an email mailing list to all psychiatrists registered with the RCPcyshiS CaAF, and subsequently, questionnaires were distributed at the RCPsychiS CaAF Conference 2023, which took place in Edinburgh, Scotland. Forms were distributed via a quick response (QR) code widely available at the conference which linked to the online form. Paper forms were also distributed. The introductory paragraphs to the survey contained an explanation for its rationale, followed by a clear statement verifying that the identity of clinicians submitting responses to the survey would remain anonymous, and that collated results would be distributed via appropriate forums including presentation and/or publication. Participants had the option to decline to be surveyed if they disagreed with this, and by commencing the survey, informed consent was confirmed by participants. There was no request for information regarding individual patients. Survey responses were received between seventh and 23 November 2023. As this was an opinion questionnaire of NHS staff members, the West of Scotland Research Ethics Service confirmed that no ethical approval was required.

### Analysis

Survey responses were described using means and standard deviations. Categorical responses were summarised using percentages. Open-ended responses regarding barriers and facilitators to clozapine use were analysed using thematic analysis, where data was coded and subsequently grouped into various themes and subthemes, with two authors (GW and JL) deriving a common consensus [10].

## Results

### Sample characteristics

Two hundred and nine eligible individuals were on the RCPsychiS CaAF mailing list and were thus sent the online survey link. The grades included in the mailing list were consultant (n = 125), higher trainee (n = 29), speciality doctor (n = 30) and retired psychiatrist (n = 25). 65 individuals attended the RCPsychiS Edinburgh conference and thus had access to the QR code and paper forms. 33 individuals accessed the online version of the survey and 30 completed the survey. Four further individuals completed a paper version. Our response rate to the survey was 34/209 relevant RCPsychiS CaAF members (16.3% of the total distribution). There are 68 practicing child and adolescent psychiatry consultants in Scotland, so our response rate is 23/68 of such professionals (33.8%) [11]. Respondent characteristics are displayed in Table 1.

**Table 1. Shows survey respondents by current grade, years of psychiatry experience and area of practice as expressed by number and percentage of total respondents.**

| Respondent characteristics | n | % of overall survey respondents | % of respondents from total members of RCPsych CaAF |
|---|---|---|---|
| **Current grade** | | | |
| Consultant | 23 | 67.6% | 18.4% |
| Higher trainee | 8 | 23.5% | 27.6% |
| Speciality doctor | 2 | 5.9% | 6.7% |
| Retired | 1 | 2.9% | 4% |
| **Years of psychiatry experience** | | | |
| 0–5 years | 2 | 5.9% | |
| 6–10 years | 9 | 26.5% | |
| 11–19 years | 11 | 32.4% | |
| 20–29 years | 9 | 26.5% | |
| 30+ years | 3 | 8.8% | |
| **Area of psychiatry worked within** | | | |
| Inpatient (Tier 4) | 5 | 14.7% | |
| Outpatient (Tier 4) | 1 | 2.9% | |
| Outpatient (Tier 3) | 14 | 41.2% | |
| Combine outpatient tier 3 and 4 | 8 | 23.5% | |
| Combined inpatient/ outpatient | 3 | 8.8% | |
| Other | 3 | 8.8% | |

## Quantitative results

**Last time of clozapine prescription.** 32.4% of respondents had prescribed clozapine to a young person in the past five years (n = 11). 44.1% of respondents had never prescribed clozapine to a young person (n = 15) (see Table 2).

**Barriers to clozapine prescription.** The most important perceived barrier according to respondents was 'concerns about compliance with clozapine monitoring' (mean = 3.3, SD = 1.1), followed by 'uncertainty if local service design allows for the completion of monitoring' (mean = 3.0, SD = 1.6) and 'organisational/ resource issues' (mean = 3.0, SD = 1.5). The barrier that was perceived to be the least important was 'uncertainty about where to get advice and support about Clozapine prescribing' (mean = 2.1, SD = 1.3) (see Fig 1).

**Perceived skill levels.** Respondents felt least skilled at 'providing medication information prior to prescription' (mean = 1.9, SD = 1.0) followed by 'identifying treatment resistant psychosis' (mean = 2.1, SD = 1.1) and 'monitoring mental health on clozapine' (mean = 2.1, SD = 1.2). Respondents felt most skilled in 'safely discontinuing clozapine' (mean = 2.8, SD = 1.2) (see Fig 2).

**Table 2. Shows the number in years of when survey respondents last prescribed clozapine, as expressed by number and percentage of total respondents.**

| Last time prescribed clozapine | n | % |
|---|---|---|
| Within 12 months | 3 | 8.8% |
| 1–2 years ago | 3 | 8.8% |
| 3–5 years ago | 5 | 14.7% |
| 6–10 years ago | 7 | 20.6% |
| 11+ years ago | 1 | 2.9% |
| Never prescribed Clozapine to a young person | 15 | 44.1% |

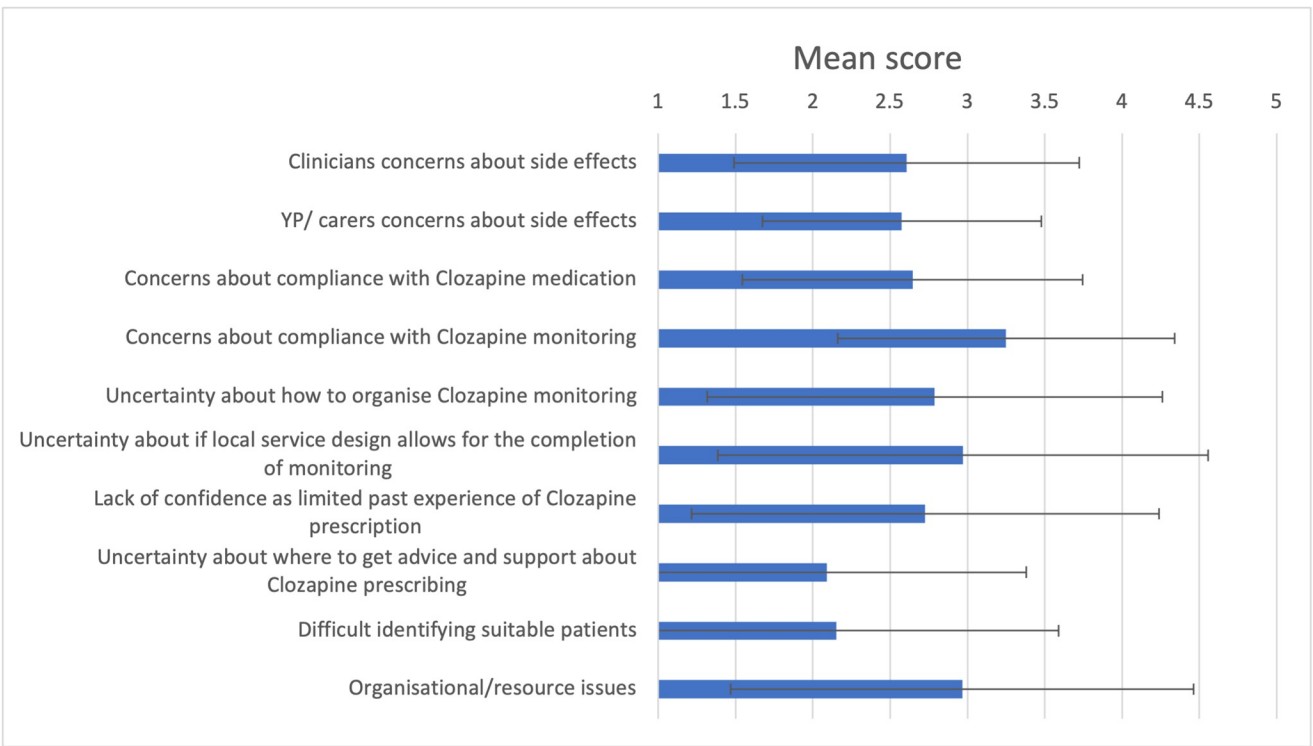

**Fig 1. Shows the mean and standard deviation of survey respondents scoring (ranging from one to five) of specific barriers to clozapine prescription.** Note one is equivalent to 'no barrier' and five is equivalent to 'major barrier'.

**Formal training accessed.** 50% of respondents had accessed NICE Guideline CG155-Psychosis and schizophrenia in children and young people (n = 17) [12]. 41.2% of respondents had accessed SIGN Guideline 131: Management of Schizophrenia (n = 14) [5]. 32.4% had attended a British Association of Psychopharmacology (BAP) course (n = 11). 35.3% reported that they had no formal training on clozapine use in young people (n = 12) (see Table 3).

**Changes that would support use of clozapine in young people.** From our quantitative questions, changes which most respondents said would support their use of clozapine in young people were improved logistics (n = 20, 58.8%), further training regarding clozapine use (n = 19, 55.9%) and raising awareness that clozapine is a treatment option in young people/ improved linkages with more experienced colleagues (n = 13, 38.2% respectively). The least popular change to support use of clozapine was improved access to clinical guidelines (eg SIGN/ NICE) (n = 3, 8.8%) (Table 4).

## Qualitative results

**Section 1: Barriers.** Three themes were identified regarding perceived barriers to clozapine use: systems and resource issues, clinician related issues, and patient related issues. The most common theme was systems and resource issues (*n* = 34), with the subthemes including lack of physical setting, blood monitoring, treatment pathways, transport, and managerial support. The second most common theme was clinician-related barriers (*n* = 17), where subthemes were prescriber confidence, medication monitoring, clinician attitudes, treatment resistant psychosis (TRP) identification, formulation, lack of qualified professionals and treatment pathways. Finally, patient-related barriers (*n* = 5) included subthemes of side effects, setting and medication monitoring. Please refer to Table 5 for a full breakdown of results.

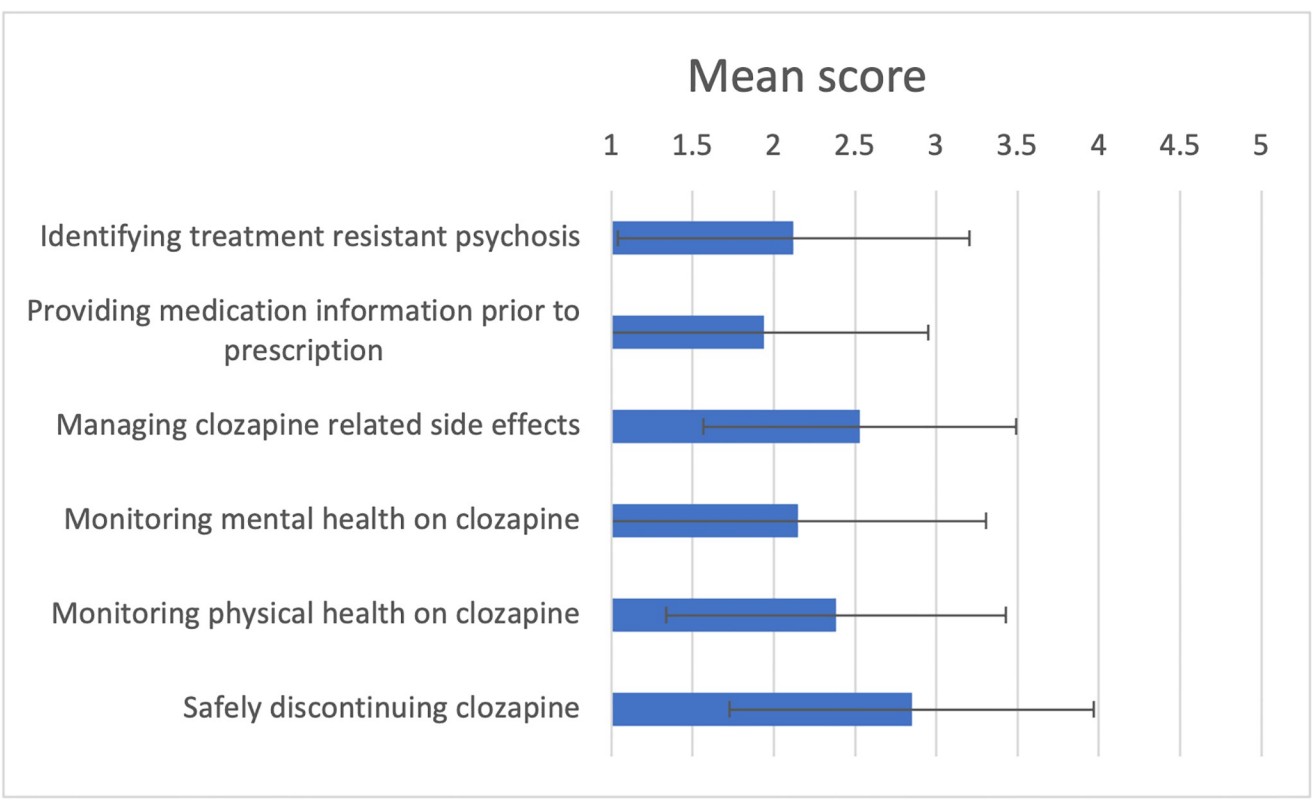

**Fig 2. Shows the mean and standard deviation of survey respondents scoring (ranging from one to five) of perceived skill levels across various areas related to clozapine prescription.** Note one is equivalent to 'unskilled' and five is equivalent to 'highly skilled'.

**Section 2: Facilitators.** Six themes were identified under this section regarding facilitators to improve clozapine prescription within CAMHS services. The most frequently mentioned themes were resources ($n = 10$), training ($n = 6$), monitoring ($n = 2$), and clinical pathways, guidance for clinicians and perspectives on clozapine treatment ($n = 1$). For resources, sub-themes were improved linkages with more experienced colleagues, increased physical spaces and increased clinical time for such cases. For training, subthemes were improved access to learning resources and raising awareness of benefits of clozapine. Please refer to Table 6 for a full breakdown of results.

## Discussion

We undertook a national survey of psychiatrists practising in CAMHS psychiatry in Scotland to examine the perceived barriers and potential facilitators to clozapine use. To the best of our

**Table 3. Shows the number of survey respondents who have completed formal clozapine related training/ CPD, with results also expressed as percentage of total respondents.**

| Formal training/ CPD accessed on clozapine use in young people | n | % |
|---|---|---|
| Accessed NICE Guideline CG155- Psychosis and schizophrenia in children and young people | 17 | 50.0% |
| Accessed SIGN Guideline 131: Management of Schizophrenia | 14 | 41.2% |
| Attended British Association of Psychopharmacology (BAP) course | 11 | 32.4% |
| No formal training | 12 | 35.3% |

**Table 4. Shows the number of survey respondents (also expressed as a percentage of total respondents) who would support various changes to support use of clozapine in young people.**

| Changes that would support use of clozapine in young people | n | % |
| --- | --- | --- |
| Improved logistics (eg access to medical room or equipment for venepuncture) | 20 | 58.8% |
| Further training in regard to Clozapine use | 19 | 55.9% |
| Improved linkages with more experienced colleagues | 13 | 38.2% |
| Raised awareness that clozapine is a treatment option in young people | 13 | 38.2% |
| Improved access to peer support groups | 8 | 23.5% |
| Improved access to pharmacy advice | 8 | 23.5% |
| Improved access to clinical guidelines (eg SIGN/ NICE) | 3 | 8.8% |

knowledge, this is the first study to have explored such issues within Scotland. This survey included the opinion of a wide range of psychiatric experience working across varying settings. 44.1% of psychiatrists had never prescribed clozapine to a young person across their entire career, with 32.4% of psychiatrists prescribing clozapine in the past five years. This result indicates that in those surveyed, clozapine is not widely utilised. This is consistent with previous research suggesting that globally, clozapine prescription is lower than recommended by gold standards [13].

Our results suggest several barriers to Clozapine use. These barriers were mainly focused on the areas of service design and resource provision. Our results suggest that for some CAMHS psychiatrists surveyed, there may be a lack of clearly defined pathways within and between services, as well as a lack of resources providing for the necessary physical monitoring of a person prescribed clozapine. Survey respondents especially commented on a perceived lack of inpatient bed availability, which would usually be required for initiation of clozapine for a young person. Globally, many experts have argued for an increase in psychiatric bed numbers [14].

Generally, respondents reported a lack of skills and confidence in clozapine initiation and follow up, with a mean score below three out of a maximum of five in all components of the perceived skills questions (Fig 2). 64.7% of respondents had accessed formal training on clozapine prescription, however we note that 32.4% of these respondents had attended a formal BAP course, and the others only reported having accessed the written NICE and SIGN guidelines. These guidelines are easily accessed with only 8.8% of respondents saying they would like access to be improved. It must be acknowledged that clozapine guidelines are not standardised across the world, and it has been suggested that harmonisation of these is recommended [15].

Almost 56% of respondents said that they would like further training regarding clozapine use, with such training focussed on raising awareness of the benefits of clozapine. A perceived lack of training opportunities has been previously in previous research as a barrier to clozapine prescription [2, 16, 17]. Such training would be important to those surveyed, however the primary facilitator to increasing clozapine prescription was around improving resources. Improved linkages with more experienced colleagues, increased clinical spaces, and time for managing such cases are indicated. Three respondents suggested that they had experienced a lack of managerial support for clozapine prescription. This issue is reflected in the recent RCPsychiS state of the nation report which suggested that clinicians and medical managers often work in silos which operate separately and share little information [18]. Medical managers in Scotland are given less time than their counterparts in England to complete service management work, hindering their ability to lead, manage and develop services [18].

**Table 5. Shows qualitative responses to questioning on barriers to clozapine use, with answers grouped into varying themes and sub-themes.** The number of respondents relating to each sub-theme is detailed, alongside the percentage of total respondents.

| Theme | Subtheme | Clinician Quotes | Subtheme mentioned by (n) | % |
|---|---|---|---|---|
| Systems and resource issues | Lack of physical setting | "difficult to access inpatient care" (n = 5) "no CAMHS clozapine clinic, so individual arrangements must be made every time", "young people attend a potentially age-inappropriate clozapine clinic set up for adults" (n = 7) "This patient's admission took much longer to arrange due to there not being an LD CAMHS adolescent unit in Scotland" | 13 | 38.2% |
| | Blood monitoring | "arrangements locally for phlebotomy significantly underestimate need for this" (n = 3) "arranging any bloods for CAMHS patients is fraught with difficulty as the clinical space, equipment and staffing to take them is lacking" (n = 3) "general practitioners will not undertake monitoring" | 7 | 20.6% |
| | Treatment Pathways | "big gap around young people coming out of hospital on clozapine in terms of local arrangements for monitoring and the burden of this falling on stretched consultants, both to create a solution without the power to alter the system, and to provide the phlebotomy" "exploring options for (blood monitoring) with primary care, paediatrics and adult mental health have not led to solutions and we have been left in a situation where CAMHS have to organise in house venepuncture" "lack of defined role of which clinic should take responsibility (CAMHS/ general adult services)" (n = 5) | 7 | 20.6% |
| | Transport | "practical issues like getting bloods to the labs" "transport likely to be an issue for patients too due to large geographical area covered" (n = 3) | 4 | 11.8% |
| | Managerial support | "far less confident that my NHS Board will be able to be flexible enough to support the required ongoing care" "managers expecting and putting relatively junior consultants/doctors in positions without support with not enough sessions to plug a gap not listening to colleagues about the service being unsafe has a part to play. If someone doesn't have the experience they shouldn't be put in that position" "when I worked in an Early Onset Psychosis service I had suggested I had the option of shadowing clinics/inpatient consultants in adult to also stay in touch with colleagues prescribing regularly but this was not supported by my managers" | 3 | 8.8% |
| Clinician related | Prescriber confidence | "I wouldn't consider prescribing off label" "lack of confidence in prescribing, lack of understanding of impact of clozapine hesitation" (n = 3) "Lack of guidance for when you would consider Clozapine use in CAMHS. Last time I prescribed it was core psychiatry training. To my knowledge it has not come up in CAMHS training (local/national teaching), BAP course or Maudsley guidelines for CAMHS. So it is difficult to know when this would be clinically appropriate to consider" (n = 2) | 6 | 17.6% |
| | Medication monitoring | "arranging routine bloods or physical monitoring difficult to arrange to a short timescale" "being able to undertake bloods and ECG can be difficult in the community" (n = 2) "I think increased clinician workload in terms of arranging prescription and physical monitoring is probably a barrier" | 4 | 11.8% |
| | Clinician attitudes | "my personal view is that CAMHS patients with psychosis should be referred to an adult consultant with experience of clozapine" "there is some discussion in patients with treatment resistant schizophrenia that clozapine might not work as well in under 18s and that we should try another antipsychotic first" | 2 | 5.9% |
| | TRP Identification | "low number of cases" "reluctance to diagnose schizophrenia" | 2 | 5.9% |
| | Formulation | "often psychotic presentations are not always differentiated as schizophrenia to follow a treatment pathway that leads to clozapine. Comorbidity with autism spectrum is common and evidence is not clear on comorbid presentation with autism in CAMHS" | 1 | 2.9% |
| | Lack of qualified professionals | "extreme shortage of CAMHS psychiatrists" | 1 | 2.9% |
| | Treatment Pathways | "lack of familiarity of processes will make this difficult if/when I need to consider" | 1 | 2.9% |

*(Continued)*

**Table 5.** (Continued)

| Theme | Subtheme | Clinician Quotes | Subtheme mentioned by (n) | % |
|---|---|---|---|---|
| Patient related | Side effects | "young people are more likely to experience side effects from antipsychotic medication and have experienced serious side effects when I have utilised clozapine in the past including cardiac arrhythmias and fever" (n = 2) | 2 | 5.9% |
| | Setting | "(inpatient) environment was unsuitable to meet needs and caused considerable disruption to services and distress to the patient (with an ID)" "issues for patients who live in more remote areas" | 2 | 5.9% |
| | Medication monitoring | "my patients in general struggle with blood tests" | 1 | 2.9% |

Previous research has looked at clinician attitudes to clozapine prescription [2, 9, 17]. The main perceived barriers to clozapine use in early intervention for psychosis services included patient concerns about side effects and physical health monitoring requirements [9]. A 2005 study involving CAMHS consultants across the United Kingdom's clozapine prescription practice concluded that unfamiliarity with the drug and the need for monitoring of side-effects seem to negatively influence use, and it might be that clearer information about effectiveness and prescribing practices would result in increased usage [2]. The need for improved information and training availability was emphasised [2]. Farooq et al conducted a systematic review

**Table 6. Shows qualitative responses to questioning on facilitators to clozapine use, with answers grouped into varying themes and sub-themes.** The number of respondents relating to each sub-theme is detailed, alongside the percentage of total respondents.

| Theme | Subtheme | Clincian Quotes | Subtheme mentioned by (n) | % |
|---|---|---|---|---|
| Resources | Improved linkages with more experienced colleagues | "If there are CAMHS colleagues very experienced with clozapine (Scotland/UK) then perhaps some CPP and peer links would help increase its use in practice." (n = 2) "more links with adult mental health, including CPD/ peer groups" (n = 5) "second opinion by adult psychiatrist confident with clozapine if young person not responding to 2 antipsychotics" | 8 | 23.5% |
| | Increased clinical spaces | "inpatient care for adolescents with learning disability" | 1 | 2.9% |
| | Increased clinical time | "(increased) time to read the guidance properly then apply it" | 1 | 2.9% |
| Training | Improved access to learning resources | "Guidance on when to prescribe in CAMHS" "I'm not sure where I'd access training" "road-shows, free online continued professional development on clozapine in young people (make sure it counts and certificate easy to get), prominent editorial (relevant journal), service users discussing their experience of clozapine in videos freely available" "there needs to be more CAMHS psychiatry influence embedded into NHS Education for Scotland to focus on this as an area of need. We therefore need to be able to access prompt high quality training" | 4 | 11.8% |
| | Raising awareness of benefits | "information on the superior benefit of clozapine in early psychosis, and the improved life expectancy of clozapine patients compared to any other antipsychotic" (n = 2) | 2 | 5.9% |
| Monitoring | Optimised monitoring | "monitoring clinics..set up for CAMHS patients" (n = 2) | 2 | 5.9% |
| Clinical pathway | TRP identification | "early psychosis pathway for Scotland that includes CAMHS is likely to help access peer support and consensus building" | 1 | 2.9% |
| Guidance for clinicians | Clozapine initiation | "easily accessible titration regimes and guidance" | 1 | 2.9% |
| Perspectives on clozapine treatment | Clinician attitudes | "change in culture and practice around diagnosing schizophrenia as I think CAMHS clinicians seem to have more of a dislike of making formal diagnosis which then results in barriers to best evidence treatment and identification of treatment-resistance." | 1 | 2.9% |

on barriers to clozapine use in an adult population [16]. The major barriers that were identified included mandatory blood testing, fear of serious side-effects and lack of patient adherence, difficulty in identifying suitable patients, service fragmentation, and inadequate training in or exposure to using clozapine [16]. A systematic review by Verdoux et al which included studies on both adults and young people, concluded that effective strategies to increase access to clozapine include implementation of integrated clozapine clinics, simplification of blood monitoring, education for prescribers and contact with experienced prescribers [17]. It was concluded that programs addressing barriers to clozapine prescription need to be disseminated more broadly. This could reduce inequalities of access to clozapine [17]. Our findings lend support to the above conclusions.

## Clinical implications

Clinicians should continue to advocate for resource provision around increasing access to appropriate settings and staffing for outpatient clinics, or inpatient beds, to commence and monitor clozapine. It is acknowledged that, in the current economic climate within the NHS, this may be challenging. Data from December 2022 shows a 14.9% WTE vacancy rate for psychiatry consultants in Scotland, compared to a rate of 6.5% for all medical specialties. Data from February 2024 shows a lower fill rate specifically for CAMHS psychiatry in Scotland, with 20.8% of full time CAMHS psychiatry consultant posts vacant [11]. The corresponding increase in requirements for clinical cover by each psychiatrist leaves less time for engagement in training and service development [18].

Ideally, time should be taken to re-evaluate service design to improve communication channels between potential clozapine prescribers and more experienced clinicians to allow knowledge to be shared more easily. Such limitations may be more prevalent in CAMHS psychiatry as opposed to adult psychiatry, due to the lower number of people being prescribed clozapine within CAMHS services. However, perhaps for clozapine prescription rates to increase within CAMHS services, resources need to be made more available and pathways clarified. When considering the principles of cost effectiveness and resource allocation, consideration could be given to non- medical professional (for example pharmacy or nursing staff) involvement in prescription and/ or monitoring of young people who are prescribed clozapine, for example via prescribing, medication explanation, monitoring or venepuncture. In the development of such services, consideration of sustainable models of care and ongoing robust and rigorous evaluation should take place [19].

Consideration should be given to specific service developments, for example point of care blood testing via finger prick, which allow a more practical, less invasive, and quicker alternative to conventional methods of monitoring clozapine levels [20]. Recent initiatives in New Zealand and the Netherlands have led to increased rates of clozapine prescription via a combination of continual audit, feedback to clinicians, educational meetings, identification of barriers to change, and the development of local guidelines for the use of clozapine [7]. Following this approach, the New Zealand clozapine prescribing rate increased from 22% to 40%, more so in areas where there were several waves of audit, local guidelines, clozapine clinics, and supervision [21].

There is currently no consistent approach to training CAMHS prescribers in Scotland around clozapine prescription and management. A more defined training pathway would be likely to increase clinician confidence [17]. In our study we detected some potential misconceptions around clozapine, for example that clozapine is not as effective in young people or should be managed by adult psychiatric services. A more consistent approach to training and

professional development could help improve prescribed knowledge and confidence around the use of clozapine in young people [17].

Such service and training development could be supported by the RCPsychiS CaAF, and in order to help shape such support, our results will be fed back to the executive committee. Our results will also be shared with the Royal College of Psychiatry Psychopharmacology Committee, who are currently developing a position statement on clozapine which will discuss expanding the prescribing of clozapine where clinically appropriate.

## Strength and limitations

To our knowledge, this is the first study to explore the perspectives of child and adolescent psychiatrists working within Scotland on barriers and facilitators to increased clozapine use. This study was distributed to all members of the RCPsychiS CaAF working with children and young people across Scotland. We posit that this survey comprised a representative sample of CAMHS psychiatrists working in Scotland as a whole, given that the number of respondents is 33.8% of the overall numbers of consultant psychiatrists working within CAMHS in Scotland.

We acknowledge the apparent low response rate; however our survey was completed by 16.3% of the total population. As with all survey-based research, there is a potential risk of bias in that a certain subgroup of clinicians with an interest in clozapine are more likely to respond. Therefore, it might be the case that knowledge and understanding of clozapine in practice is lower in the whole population, and that our results may overestimate overall levels of confidence and knowledge in the use of clozapine.

An important limitation is the absence of any service user perspective on this matter. Such data would be especially useful when considering patient related barriers to clozapine use, which are described here by the treating psychiatrist instead of directly by the service user who may have a differing perspective. Finally, given the relatively small workforce numbers, and to keep results anonymous, we did not ask where respondents worked. Thus, through our findings we cannot make comment on whether barriers and facilitators are more prevalent in specific areas, nor suggest which may be the best regions to target with service improvement methodologies to improve clozapine prescription rates.

## Future research suggestions

Our research gives a snapshot of the viewpoint of child and adolescent psychiatrists across Scotland. Future studies could utilise more in-depth qualitative methods using an interview or focus group approach. CAMHS services vary across the country, and thus perhaps a more health board specific approach to look at local barriers and facilitators for improvement would be useful. Future research should consider the views of service users and their families/ carers on barriers and facilitators to improve clozapine treatment, for example via a similar targeted questionnaire or other qualitative methods. Future research should aim for improved response rates, considering methodologies such as modest rewards for participation, for example.

## Conclusion

Our results suggest that the main perceived barriers to clozapine use in CAMHS services within Scotland were systems and resource issues. The main facilitators to improving clozapine prescription were seen to be improving resources and training opportunities. National policymakers should provide adequate funding for such facilitators, to help reduce barriers and potentially improve the care of young people with schizophrenia in Scotland. National policy influencers including the Mental Welfare Commission, NHS Education for Scotland, and NHS Scotland should consider these findings to address the potential underutilisation of

clozapine for people aged under 18 in services across Scotland. This may help to improve outcomes for young people who experience treatment-resistant psychosis.

## Acknowledgments

The authors would like to acknowledge the members of the RCPsychiS CaAF executive committee, who contributed to the development of this study, as well as all members of the RCPsychiS faculty who contributed to our results.

## Author Contributions

**Conceptualization:** Graham Walker, Helen Smith.

**Data curation:** Graham Walker.

**Formal analysis:** Graham Walker, Jason Lang.

**Investigation:** Graham Walker.

**Methodology:** Graham Walker, Helen Smith.

**Supervision:** Jason Lang.

**Writing – original draft:** Graham Walker.

**Writing – review & editing:** Graham Walker, Jason Lang, Helen Smith.

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
