## [Decision Letter · Decision Letter 0]

26 Apr 2024

PONE-D-24-07703 Survey on barriers to psychiatrists’ use of clozapine for young people in Scotland and suggestions for reducing these

PLOS ONE

Dear Dr. Walker,

Thank you for submitting your manuscript to PLOS ONE. After careful consideration, we feel that it has merit but does not fully meet PLOS ONE’s publication criteria as it currently stands. Therefore, we invite you to submit a revised version of the manuscript that addresses the points raised during the review process.

We look forward to receiving your revised manuscript.

Kind regards,

Yasin Hasan Balcioglu, MD, PhD

Action Editor

PLOS ONE

Journal Requirements:

3. Please remove your figures from within your manuscript file, leaving only the individual TIFF/EPS image files, uploaded separately. These will be automatically included in the reviewers’ PDF.

Reviewers' comments:

Reviewer's Responses to Questions

**Comments to the Author**

1. Is the manuscript technically sound, and do the data support the conclusions?

Reviewer #1: Yes

Reviewer #2: Yes

2. Has the statistical analysis been performed appropriately and rigorously? 

Reviewer #1: Yes

Reviewer #2: Yes

3. Have the authors made all data underlying the findings in their manuscript fully available?

Reviewer #1: Yes

Reviewer #2: Yes

4. Is the manuscript presented in an intelligible fashion and written in standard English?

Reviewer #1: Yes

Reviewer #2: Yes

5. Review Comments to the Author

Reviewer #1: Dear Author/s

Thank you for your study. This study is about the perspective of experts working with children and adolescents in Scotland on the use of clozapine. It gives the impression of an original survey with detailed questions on the subject. Although the results are discussed in detail in the discussion section, they can be further supported by studies conducted from different countries on clozapine use problems, even in adulthood.

Best regards.

Reviewer #2: I read the article titled Survey on barriers to psychiatrists' use of clozapine for young people in Scotland and suggestions for reducing these, and in general, I think the article is of high quality. Doing it with Mixst methodology made it easier to verify the hypotheses. My suggestions are as follows.

1) In the tables in the study, the question with the highest percentage should be at the top, loaded tables make it difficult to read. At the same time, quotes can be slightly reduced or summarized.

6. PLOS authors have the option to publish the peer review history of their article (what does this mean?). If published, this will include your full peer review and any attached files.

Reviewer #1: **Yes: **Cansu Çobanoğlu Osmanlı

Reviewer #2: **Yes: **Senay Kilincel

---

## [Author Response · Author response to Decision Letter 0]

1 May 2024

Dear Dr Balcioglu,

Thank you for providing your reviewers comments on our manuscript entitled “Survey on barriers to psychiatrists use of clozapine for young people in Scotland and suggestions for reducing these”.

I can confirm we have made the following recommended changes to our manuscript;

• We have updated our manuscript as per the PLOS One formatting guidelines.

• We have moved our ethics statement into the recruitment section of our methods as directed.

• We have removed figures from the main manuscript as directed.

• Our reference list has been updated with new included papers, focusing on relating our results to previous research.

• In response to reviewer 1, we have added several other references to clozapine related studies, including those considering a more global perspective.

• In response to reviewer 2, we have changed the order or the qualitative feedback tables to highest percentage at the top as advised. We have changed the corresponding themes/ subthemes in the text to this same order. We have also summarised the quotes to a more succinct level.

• We have uploaded our two figure files to the Preflight Analysis and Conversion Engine (PACE) digital diagnostic tool.

We hope that the following changes will make this article suitable for publication in your journal. If you would recommend any further changes, we are very happy to consider these also.

Yours sincerely,

Dr Graham Walker

BSc (Med Sci), MBChB, MRCPsych

Clinical Lecturer in Child and Adolescent Psychiatry, The University of Glasgow

ST4 CAMHS Psychiatry Registrar, West of Scotland

---

## [Editor Report · Decision Letter 1]

22 May 2024

Survey on barriers to psychiatrists’ use of clozapine for young people in Scotland and suggestions for reducing these

PONE-D-24-07703R1

Dear Dr. Walker,

We’re pleased to inform you that your manuscript has been judged scientifically suitable for publication and will be formally accepted for publication once it meets all outstanding technical requirements.

Kind regards,

Yasin Hasan Balcioglu, MD, PhD

Action Editor

PLOS ONE
---

## [Editor Report · Acceptance letter]

27 May 2024

PONE-D-24-07703R1 

PLOS ONE

Dear Dr. Walker, 

I'm pleased to inform you that your manuscript has been deemed suitable for publication in PLOS ONE. Congratulations! Your manuscript is now being handed over to our production team.

Kind regards, 

on behalf of

Dr. Yasin Hasan Balcioglu 

Academic Editor

PLOS ONE